# Modes of HIV transmission among young women and their sexual partners in Ukraine

**Oleksandr Zeziulin** [1] *, **Maryna Kornilova** [2], **Alexandra Deac** [3], **Olga Morozova** [4], **Olga Varetska** [2], **Iryna Pykalo** [1], **Kostyantyn Dumchev** [1]

**1** European Institute of Public Health Policy, Kyiv, Ukraine, **2** The International Charitable Foundation "Alliance for Public Health", Kyiv, Ukraine, **3** Department of Health Service and Population Research, King's College London, London, United Kingdom, **4** Department of Public Health Sciences, University of Chicago, Chicago, IL, United States of America

* zeziulin@uiphp.org.ua

## Abstract

### Background

Ukraine has the second-largest HIV epidemic in Europe, with most new cases officially attributed to heterosexual transmission. Indirect evidence suggested substantial HIV transmission from people who inject drugs (PWID) to their sexual partners. This study examined the extent of heterosexual HIV transmission between PWID and non-drug-using adolescent girls and young women (AGYW).

### Methods

A cross-sectional survey recruited AGYW diagnosed with heterosexually-acquired HIV between 2016 and 2019 in nine regions of Ukraine. AGYW were asked to identify and refer their sexual partners ('Partners'), who subsequently underwent HIV testing, and, if positive, HCV testing. Both AGYW and Partners completed an interview assessing HIV risk behaviors prior to AGYW's HIV diagnosis.

### Results

In August-December 2020, we enrolled 321 AGYW and 64 Partners. Among the Partners, 42% either self-reported IDU or were HCV-positive, indicating an IDU-related mode of HIV transmission. PWID Partners were more likely to report sexually transmitted infections (STI) and had lower educational levels. Of the 62 women who recruited at least one Partner, 40% had a PWID Partner. Within this subgroup, there was a higher prevalence of STIs (52% vs. 24%) and intimate partner violence (36% vs. 3%). Condom use was less common (52% vs. 38% reporting never use), and frequent alcohol or substance use before sex was higher (48% vs 30%) among AGYW with PWID Partner, although this difference did not reach statistical significance. Notably, 52% of women were aware of their Partners' IDU.

### Conclusion

At least 40% of heterosexual transmission among AGYW in Ukraine can be linked to PWID partners. Intensified, targeted HIV prevention efforts are essential for key and bridge

**Data Availability Statement:** All relevant data are within the manuscript and its Supporting Information files.

**Funding:** The study was conducted under the financial support of International Charitable Foundation "Public Health Alliance" (hereinafter the Alliance) which is a leading professional organization that, in cooperation with key public organizations, the Ministry of Health and other government bodies of Ukraine, fights the HIV/AIDS epidemic in Ukraine, managing preventive programs and providing quality technical support and financial resources to organizations. The Alliance experts contributed to the study design and data analysis, decision to publish, and preparation of the manuscript.

**Competing interests:** The authors have declared that no competing interests exist.

populations (PWID and their sexual partners), addressing the biological and structural determinants of transmission between key and bridge populations, such as IDU- and HIV status disclosure, STIs, IPV, and stigma.

## Introduction

Despite the substantial progress in the fight against HIV/AIDS over the past three decades [1], people who inject drugs (PWID) and their sexual partners remain vulnerable, facing increased risks of contracting and transmitting HIV [2]. Current evidence suggests that HIV key populations, including PWID, and their sexual partners account for up to 65% of all new HIV infections worldwide [3]. PWID often engage in risky sexual practices, such as having unprotected sex or having multiple sexual partners, who are often non-PWID [4,5].

Ukraine has the second largest HIV epidemic in Europe, with an estimated 240,000 people living with HIV in 2020 [6]. Official case registration data indicate that heterosexual transmission has been the dominant mode in Ukraine since 2008 [7]. However, epidemiological investigations have revealed significant misclassification of transmission modes [8,9], with a subsequent survey suggesting that at least 55% of HIV cases registered between 2013 and 2015 were likely attributed to injecting drug use (IDU) [10]. From 2015 until 2019, the distribution of primary transmission categories remained relatively stable, confirming sustained transmission levels among PWID, and emphasizing the continued significance of heterosexual mode. Indirect evidence was indicating that heterosexual transmission among women in Ukraine remained primarily linked to PWID partners [11–13], which may be particularly true for young women who do not inject drugs [14]. This evidence has guided Ukraine's National HIV Program to maintain its focus on prevention programs targeting key populations (PWID, men who have sex with men, and commercial sex workers) as well as bridge populations (sexual partners of key populations) [15].

Adolescent girls and young women (AGYW) are at a higher risk of HIV transmission due to early sexual debut, unprotected sex, and having sexual partners who engage in high-risk behaviors such as drug use, underscoring prominent gender differences in contracting HIV [16,17]. Heterosexual AGYW face disproportional and multifaceted challenges in accessing HIV prevention programs, and barriers to HIV disclosure, testing and care due to psychosocial pressure and stigma [18–20]. According to Spectrum model, approximately 2,200–3,500 AGYW lived with HIV in Ukraine in 2019 [6].

Despite these emerging concerns, significant gaps persist in epidemiological knowledge about the specific risk factors and patterns of HIV transmission from PWID to AGYW who do not inject drugs [21]. Given the lack of knowledge, this potentially important bridge population was mostly neglected by HIV prevention programs in Ukraine. This study aimed to investigate the extent to which heterosexually acquired HIV cases among AGYW can be linked to their sexual partners who injected drugs. These results will help to understand whether transmission beyond key and bridge populations is substantial and whether there is a risk of HIV epidemic generalization, providing pivotal information for prevention efforts.

## Methods

### Study population and recruitment

We adapted the methodology from the previous study to enroll a random sample of women living with HIV and assess their HIV risk factors [10]. We employed the index testing

approach [22] to recruit and assess their sexual partners. The primary target population was women diagnosed with HIV between the ages of 15–25 years in 2016–2019, who were registered with heterosexual mode of transmission. To obtain nationally representative estimates, the study was conducted in nine (of 27) geographically and epidemiologically diverse regions of the country: 1) Northern regions: Kyiv city and Kyiv Oblast; 2) Eastern regions: Dnipro, Donetsk; 3) Southern regions: Odesa, Mykolaiv 4) Central regions: Cherkasy, Zhytomyr 5) Western regions: Lviv, Volyn. The target sample size was distributed across these regions proportionally to the total number of HIV cases registered between 2016 and 2019.

Clinical staff at participating HIV clinics utilized the electronic HIV Medical Information System (MIS) to extract lists of IDs for women meeting the target population definition, stratified by year of diagnosis. Our study staff then performed random selection of the required number of patient IDs to achieve the target sample size (adjusted for non-response) for each site, and populated recruitment logs. Clinic staff used available information to contact the potential participants from the logs via phone or during clinical visits, read a standard invitation script, and scheduled study appointments at the clinic. After obtaining informed consent, participants were questioned about their history of IDU prior to HIV diagnosis and were tested for anti-HCV antibodies if previous test information was not available in the medical chart. Women who reported IDU or were HCV-positive were excluded from the study, as they were considered to have acquired HIV via parenteral mode.

The secondary study population was men who could potentially transmit HIV through sexual contact to participating AGYW. AGYW were requested to identify up to five men with whom they had sexual contacts before their HIV diagnosis and who could be a potential source of HIV infection ('Partners'). For each Partner, women responded to three questions about history of intimate partner violence (IPV) or abuse (S1 Table). Those who met at least one 'unsafe' criterion were excluded from enrollment. For eligible Partners, women could choose either to refer them themselves using coupons or opt for referral by study staff. The process of status disclosure (if necessary) and referral was conducted according to the index testing guidelines [22]. Partners who arrived at the study site were screened to match the visual description provided by the referee. If matched, they were asked about HIV status. Those who knew about their status were verified using MIS, and those with negative or unknown status were tested for HIV. Those testing negative were excluded from the study. Men who tested positive for the first time or were not registered in HIV care were referred to HIV clinic staff for counseling and appropriate services, and later returned to the study.

### Data collection

Both participant groups completed a self-administered assisted survey using REDCap electronic data capture tools [23] hosted at European Institute of Public Health Policy (Kyiv, Ukraine). The questionnaire assessed HIV risk factors that took place before the AGYW HIV diagnosis (including sexual behavior, drug and alcohol use, sexually transmitted infections (STI) history, parenteral exposures), self-reported way of HIV acquisition, and demographical information. We also collected data from MIS on the registered mode of HIV transmission and HCV status of the Partners. Partners who did not have data on HCV in their medical record, were tested for anti-HCV antibodies before the survey using a rapid test kit available at the clinic.

### Data analysis

Frequencies and proportions for sociodemographic and behavioral variables were used to characterize the sampled participants separately for AGYW and Partners. We used logical

formulas for risk behavior definitions of HIV risk factors (heterosexual, homosexual, injecting drug use, nosocomial, accidental and sexually transmitted infections; S2 Table). These variables were treated as not mutually exclusive, recognizing that individuals might be exposed to more than one factor simultaneously. In Partners, anti-HCV positivity was considered a marker of IDU exposure due the high parenteral transmissibility [24], low likelihood of sexual transmission [25], and relatively low prevalence in general population in Ukraine [26].

For Partners, we constructed a summary variable representing the most probable mode of transmission based on the survey responses (survey-based mode of HIV transmission, SMoT), based on the previous study approach [10]. If IDU risk factor was present (self-reported or assumed due to HCV positivity), SMoT was assigned as IDU. Men who had no IDU but reported male-to-male sex, were considered to be infected through MSM exposure. Others were assigned heterosexual SMoT.

We compared AGYW who had a Partner with IDU SMoT with those who did not. The differences in socio-demographic and HIV risk characteristics were assessed using chi-square and Fischer's exact test for subtables with expected counts less than 5. A similar approach was used to compare Partners with or without IDU SMoT. To assess the extent of selection bias, we conducted a sensitivity analysis comparing women who had at least one Partner recruited to those who had not.

Statistical analyses were done using R version 4.3.2 [27].

### Ethical statement

The Institutional Review Board (IRB) approved the study protocol at the Ukrainian Institute on Public Health Policy (Kyiv, Ukraine). Approval number: #28-20/IRB. All data collected were kept secure and accessible only to authorized research team members. Strict adherence to confidentiality was kept at all times. Written informed consent was obtained from all study participants. All participants were compensated for time spent on the study.

### Results

A total of 321 AGYW with a likely heterosexual mode of HIV transmission were enrolled during the study enrollment period from August to December 2020 (Table 1). Median age at the time of diagnosis was 23 (IQR 21–24) years old, while median age at the time of the survey was 25 (IQR 23–27). Socio-demographic characteristics and HIV risk factors of AGYW are presented in Table 1.

Nearly 60% of women reported sexual contacts with men living with HIV before finding out about their own HIV status, and 19% reported having sex with men who injected drugs according to their knowledge. Only 26% used condoms regularly, and 33% often used alcohol or drugs before sex. Selling sex for money was reported by 3% of women, and 27% had at least one sexually transmitted infection.

Nearly all (314/321, 98%) of women were able to name and describe at least one partner who could be a potential source of heterosexual HIV transmission, 83% had at least one 'safe' partner without IPV or abuse history, and 38% agreed to refer partners for the study (S1 Table). A total of 66 Partners of 62 AGYW were successfully recruited, two tested negative for HIV and were excluded.

Most Partners were about 30 years old at the time of the survey (median 30, IQR 27–33). Fifty-one (80%) were already registered in HIV clinic with median time in care of 38 months (IQR 22–51); others tested positive during the survey. Sixteen (31%) were registered with IDU MoT, and 21 (33%) tested positive for anti-HCV (Table 2). In the survey, 25 (39%) self-reported any IDU history, and 2 (3%) reported male-to-male sex. Combined, these variables

**Table 1. Socio-demographic characteristics and HIV risk factors of AGYW in the study sample.**

|  |  | N | % |
|---|---|---|---|
| Total |  | 321 | 100.0 |
| Age at the survey | <20 | 12 | 3.7 |
|  | 20–24 | 142 | 44.2 |
|  | 25+ | 167 | 52.0 |
| Age at HIV registration | <20 | 62 | 19.3 |
|  | 20–24 | 228 | 71.0 |
|  | 25 | 31 | 9.7 |
| Education | school | 180 | 56.1 |
|  | technical | 92 | 28.7 |
|  | higher | 49 | 15.3 |
| Employment | employed | 166 | 51.7 |
|  | unemployed | 70 | 21.8 |
|  | student | 59 | 18.4 |
|  | other | 26 | 8.1 |
| Family status | single | 229 | 71.3 |
|  | married | 85 | 26.5 |
|  | separated | 7 | 2.2 |
| Time from testing to registration | <3 months | 196 | 81.7 |
|  | 3–11.99 months | 25 | 10.4 |
|  | 12+ months | 19 | 7.9 |
| Time in care | <1 year | 28 | 8.7 |
|  | 1–1.99 years | 76 | 23.7 |
|  | 2–2.99 years | 85 | 26.5 |
|  | 3+ years | 132 | 41.1 |
| Condom use | never/rarely | 129 | 40.2 |
|  | 50/50 | 108 | 33.6 |
|  | often/always | 84 | 26.2 |
| Alcohol/substance use before sex | never | 50 | 15.6 |
|  | sometimes | 165 | 51.4 |
|  | often/always | 106 | 33.0 |
| Attended places where others used drugs | no | 295 | 91.9 |
|  | yes | 26 | 8.1 |
| STI history | no | 233 | 72.6 |
|  | yes | 88 | 27.4 |
| Nosocomial exposure | no | 188 | 58.6 |
|  | yes | 133 | 41.4 |
| Accidental exposure | no | 250 | 77.9 |
|  | yes | 71 | 22.1 |
| Sex with male PWID | no | 259 | 80.7 |
|  | yes | 62 | 19.3 |
| Sex with MSM | no | 318 | 99.1 |
|  | yes | 3 | 0.9 |
| Sex with male PLWH | no | 133 | 41.4 |
|  | yes | 188 | 58.6 |
| Sex with male sex worker | no | 314 | 97.8 |
|  | yes | 7 | 2.2 |
| Selling sex for money | no | 311 | 96.9 |

(*Continued*)

**Table 1.** (Continued)

|  |  | N | % |
|---|---|---|---|
|  | yes | 10 | 3.1 |
| History of IPV with named partners | no | 280 | 87.2 |
|  | yes | 41 | 12.8 |

AGYW, adolescent girls and young women; STI, sexually transmitted infection; PWID, people who inject drugs; MSM, men who have sex with men; PLWH, people living with HIV, IPV, intimate partner violence.

resulted in 27 (42%) of Partners assigned with IDU, 1 (2%) with MSM, and 36 (56%) with heterosexual SMoT. Among those already registered in care, the proportion of IDU SMoT was 45% (23/51). In comparison, Partners with IDU SMoT were less educated, had higher probability of nosocomial and accidental exposure, more likely had sex with women who injected drugs, and much higher frequency of STI history (Table 2).

Among the 62 women who recruited at least one Partner, 25 (40.3%) had a Partner who likely acquired HIV through IDU (Table 3). AGYW who had a recruited Partner with IDU MoT exhibited a significantly higher frequency of STI (52% compared to 24% among those with non-IDU partners, p = 0.05). Notably, only 52% of AGYW with a IDU MoT Partner reported having sex with PWID, indicating low awareness of women about injecting behavior of their partners.

Women with an IDU MoT Partner were much more likely to report IPV with either of the named partners (36% vs 3%, p = 0.001). Sexual contacts with men with known HIV status was very high in both groups (88% and 70%), although the difference was not significant. Sex with MSM, and buying sex from a male sex worker was rare and did not differ significantly. Selling sex for money was reported on average by 11% and did not differ between subgroups. Condom use was lower (52% vs 38% never used), and frequent alcohol or substance use before sex was more prevalent (48% vs 30%) among AGYW with an IDU MoT Partner, but these differences did not reach statistical significance.

The sensitivity analysis, comparing women who had a recruited Partner in the study (N = 62) with those who had not (N = 259), revealed that the groups were similar (S3 Table). No differences were observed in socio-demographic characteristics or risk practices. STI history was higher among those who had a Partner (36% vs 26%), but the difference was not significant. Women with PWID Partners were more likely to engage in sex sork (11% vs 1%, p = 0.001). Reports of IPV were similar between the two groups, with 16% and 12% respectively, without statistical difference. Importantly, sexual contacts with men living with HIV was 77% among AGYW with a Partner and 54% among those without one (p = 0.001), while any sex with male PWID was 19% on average without a notable difference between the subgroups.

## Discussion

In this study we explored the extent to which heterosexual HIV transmission among non-PWID adolescent girls and young women in Ukraine is linked to the epidemic among PWID. Our approach involved recruiting sexual partners of AGYW and assessing their probable mode of transmission. The key findings was that at least 40% of heterosexually-infected AGYW could acquire HIV from PWID sexual partners, suggesting that the complex interplay between drug use and sexual behavior in key and bridge populations continues to be the driving force of the HIV epidemic in Ukraine.

**Table 2. Socio-demographic characteristics and HIV risk factors of Partners, by SMoT.**

| | | | | SMoT | | | | | |
|---|---|---|---|---|---|---|---|---|---|
| | | Total | | IDU | | Not IDU | | Chi-sq. | p-value |
| | | N | Col. % | N | Col. % | N | Col. % | | |
| Total (Row %) | | 64 | 100.0 | 27 | 42.2 | 37 | 57.8 | | |
| Age at the survey | <30 | 27 | 44.3 | 11 | 44.0 | 16 | 44.4 | F | 0.794 |
| | 30–34 | 22 | 36.1 | 10 | 40.0 | 12 | 33.3 | | |
| | 35+ | 12 | 19.7 | 4 | 16.0 | 8 | 22.2 | | |
| Age at HIV registration | <25 | 17 | 35.4 | 7 | 33.3 | 10 | 37.0 | F | 0.809 |
| | 25–29 | 20 | 41.7 | 10 | 47.6 | 10 | 37.0 | | |
| | 30+ | 11 | 22.9 | 4 | 19.0 | 7 | 25.9 | | |
| Education | school | 24 | 37.5 | 13 | 48.1 | 11 | 29.7 | 7.82 | **0.020** |
| | technical | 27 | 42.2 | 6 | 22.2 | 21 | 56.8 | | |
| | higher | 13 | 20.3 | 8 | 29.6 | 5 | 13.5 | | |
| Employment | employed | 55 | 85.9 | 22 | 81.5 | 33 | 89.2 | F | 0.636 |
| | unemployed | 6 | 9.4 | 3 | 11.1 | 3 | 8.1 | | |
| | other | 3 | 4.7 | 2 | 7.4 | 1 | 2.7 | | |
| Family status | single | 44 | 68.8 | 19 | 70.4 | 25 | 67.6 | F | 0.380 |
| | married | 14 | 21.9 | 7 | 25.9 | 7 | 18.9 | | |
| | separated | 6 | 9.4 | 1 | 3.7 | 5 | 13.5 | | |
| Registered in HIV clinic | no | 13 | 20.3 | 4 | 14.8 | 9 | 24.3 | 0.38 | 0.536 |
| | yes | 51 | 79.7 | 23 | 85.2 | 28 | 75.7 | | |
| Time from testing to registration | <3 months | 38 | 77.6 | 16 | 72.7 | 22 | 81.5 | F | 0.784 |
| | 3–11.99 months | 7 | 14.3 | 4 | 18.2 | 3 | 11.1 | | |
| | 12+ months | 4 | 8.2 | 2 | 9.1 | 2 | 7.4 | | |
| Time in care | <2 years | 14 | 27.5 | 7 | 30.4 | 7 | 25.0 | 0.71 | 0.702 |
| | 2–3.99 years | 21 | 41.2 | 8 | 34.8 | 13 | 46.4 | | |
| | 4+ years | 16 | 31.4 | 8 | 34.8 | 8 | 28.6 | | |
| Registered MoT | IDU | 16 | 31.4 | 16 | 69.6 | | | | |
| | heterosexual | 35 | 68.6 | 7 | 30.4 | 28 | 100.0 | | |
| HCV test result | neg | 43 | 67.2 | 6 | 22.2 | 37 | 100.0 | | |
| | pos | 21 | 32.8 | 21 | 77.8 | | | | |
| Self-reported IDU exposure | no | 39 | 60.9 | 2 | 7.4 | 37 | 100.0 | | |
| | yes | 25 | 39.1 | 25 | 92.6 | | | | |
| Homosexual exposure | no | 62 | 96.9 | 26 | 96.3 | 36 | 97.3 | F | 1.000 |
| | yes | 2 | 3.1 | 1 | 3.7 | 1 | 2.7 | | |
| STI history | no | 56 | 87.5 | 20 | 74.1 | 36 | 97.3 | F | **0.008** |
| | yes | 8 | 12.5 | 7 | 25.9 | 1 | 2.7 | | |
| Nosocomial exposure | no | 35 | 54.7 | 9 | 33.3 | 26 | 70.3 | 7.17 | **0.007** |
| | yes | 29 | 45.3 | 18 | 66.7 | 11 | 29.7 | | |
| Accidental exposure | no | 41 | 64.1 | 9 | 33.3 | 32 | 86.5 | 16.92 | **0.000** |
| | yes | 23 | 35.9 | 18 | 66.7 | 5 | 13.5 | | |
| Survey-based MoT | heterosexual | 36 | 56.3 | | | 36 | 97.3 | | |
| | IDU | 27 | 42.2 | 27 | 100.0 | | | | |
| | MSM | 1 | 1.6 | | | 1 | 2.7 | | |
| Sex with a woman who injects drugs | no | 55 | 85.9 | 19 | 70.4 | 36 | 97.3 | F | **0.003** |
| | yes | 9 | 14.1 | 8 | 29.6 | 1 | 2.7 | | |
| Sex with a woman living with HIV | no | 38 | 59.4 | 19 | 70.4 | 19 | 51.4 | 1.62 | 0.203 |
| | yes | 26 | 40.6 | 8 | 29.6 | 18 | 48.6 | | |

(*Continued*)

**Table 2.** (Continued)

| | | Total | | SMoT IDU | | SMoT Not IDU | | Chi-sq. | p-value |
|---|---|---|---|---|---|---|---|---|---|
| | | N | Col. % | N | Col. % | N | Col. % | | |
| Buying sex from women | no | 46 | 71.9 | 22 | 81.5 | 24 | 64.9 | 1.39 | 0.239 |
| | yes | 18 | 28.1 | 5 | 18.5 | 13 | 35.1 | | |
| Selling sex | no | 64 | 100.0 | 27 | 100.0 | 37 | 100.0 | | |
| Condom use | never/rarely | 33 | 51.6 | 13 | 48.1 | 20 | 54.1 | 4.05 | 0.132 |
| | 50/50 | 17 | 26.6 | 5 | 18.5 | 12 | 32.4 | | |
| | often/always | 14 | 21.9 | 9 | 33.3 | 5 | 13.5 | | |
| Alcohol/substance use before sex | never/sometimes | 18 | 28.1 | 8 | 29.6 | 10 | 27.0 | 0.00 | 1.000 |
| | often/always | 46 | 71.9 | 19 | 70.4 | 27 | 73.0 | | |

SMoT, survey-based mode of HIV transmission; MoT, mode of HIV transmission; IDU, injecting drug use; STI, sexually transmitted infection; MSM, men who have sex with men.

F denotes that Fischer's exact test was used instead of chi-square test.

Importantly, only about half or women with a Partner likely infected via IDU were aware of their partner's injecting behavior. This may have important implications for personal risk perception and condom use [28], which should be taken into account in prevention programs targeting this bridge population. Additionally, this highlights that self-reported information from women about drug use of their partners may not accurately reflect their belonging to the bridge population.

Our results indicate that having a PWID Partner was associated with a higher frequency of STI, thereby increasing the probability of acquiring HIV through sexual contact [29]. Other factors that may further impact vulnerability to HIV, namely serodiscordant sexual relationships and condomless sex were also more prevalent among AGYW-partners of PWID, although the study did not have sufficient power to reach significance. These findings alight with existing evidence demonstrating elevated HIV risk among sexual partners of PWID in various cultural and economic contexts [30,31]. Transmission of HIV and other STIs from PWID to AGYW is substantially increased by psychosocial and structural factors, including unprotected sex, alcohol consumption, drug use, homelessness, stigma, and lack of awareness [32,33]. The importance of these factors is also confirmed in our comparison of PWID and non-PWID Partners, showing that the former group had a lower educational level and a higher risk of STIs.

Another important finding was that while 13% of all AGYW reported history of IPV with partners who could transmit HIV to them, women having a PWID Partner were twelve times more likely to report IPV than those with non-PWID sexual partners (36% compared to 3%). Worldwide IPV prevalence reaches up to 20% among women and varies across types of IPV (physical, psychological, and/or sexual), geographical locations and measurement purposes [34]. Although HIV-positive women appear to experience IPV at rates comparable to HIV-negative women from the same underlying populations, their abuse seems more frequent and severe [35]. This result highlights the multidimensional risk factors of heterosexual HIV transmission and underscores the need for integration of HIV prevention and IPV interventions among sexual partners of PWID [36].

As a secondary objective, we were able to assess the magnitude of misclassification of IDU MoT among Partners. The proportion of cases attributed to IDU was 31% in the official registration record, compared to 45% according to self-report in the survey combined with anti-

**Table 3. Comparison of socio-demographic characteristics and HIV risk factors of AGYW, by Partners' SMoT.**

| | | Total | | IDU | | Not IDU | | Chi-sq. | p-value |
|---|---|---|---|---|---|---|---|---|---|
| | | N | Col. % | N | Col. % | N | Col. % | | |
| Total (Row %) | | 62 | 100.0 | 25 | 40.3 | 37 | 59.7 | | |
| Age at the survey | <20 | 1 | 1.6 | 1 | 4.0 | | | F | 0.418 |
| | 20–24 | 26 | 41.9 | 9 | 36.0 | 17 | 45.9 | | |
| | 25+ | 35 | 56.5 | 15 | 60.0 | 20 | 54.1 | | |
| Age at HIV registration | <20 | 14 | 22.6 | 5 | 20.0 | 9 | 24.3 | F | 0.116 |
| | 20–24 | 42 | 67.7 | 15 | 60.0 | 27 | 73.0 | | |
| | 25 | 6 | 9.7 | 5 | 20.0 | 1 | 2.7 | | |
| Education | school | 36 | 58.1 | 17 | 68.0 | 19 | 51.4 | F | 0.319 |
| | technical | 20 | 32.3 | 7 | 28.0 | 13 | 35.1 | | |
| | higher | 6 | 9.7 | 1 | 4.0 | 5 | 13.5 | | |
| Employment | employed | 36 | 58.1 | 17 | 68.0 | 19 | 51.4 | F | 0.103 |
| | unemployed | 13 | 21.0 | 4 | 16.0 | 9 | 24.3 | | |
| | student | 6 | 9.7 | | | 6 | 16.2 | | |
| | other | 7 | 11.3 | 4 | 16.0 | 3 | 8.1 | | |
| Family status | single | 47 | 75.8 | 16 | 64.0 | 31 | 83.8 | F | 0.092 |
| | married | 14 | 22.6 | 8 | 32.0 | 6 | 16.2 | | |
| | separated | 1 | 1.6 | 1 | 4.0 | | | | |
| Time from testing to registration | <3 months | 40 | 88.9 | 7 | 87.5 | 33 | 89.2 | F | 0.643 |
| | 3–11.99 months | 2 | 4.4 | | | 2 | 5.4 | | |
| | 12+ months | 3 | 6.7 | 1 | 12.5 | 2 | 5.4 | | |
| Time in care | <1 year | 5 | 8.1 | 2 | 8.0 | 3 | 8.1 | F | 0.757 |
| | 1–1.99 years | 14 | 22.6 | 4 | 16.0 | 10 | 27.0 | | |
| | 2–2.99 years | 21 | 33.9 | 10 | 40.0 | 11 | 29.7 | | |
| | 3+ years | 22 | 35.5 | 9 | 36.0 | 13 | 35.1 | | |
| Condom use | never/rarely | 27 | 43.5 | 13 | 52.0 | 14 | 37.8 | 1.24 | 0.539 |
| | 50/50 | 21 | 33.9 | 7 | 28.0 | 14 | 37.8 | | |
| | often/always | 14 | 22.6 | 5 | 20.0 | 9 | 24.3 | | |
| Alcohol/substance use before sex | never | 7 | 11.3 | 1 | 4.0 | 6 | 16.2 | F | 0.213 |
| | sometimes | 32 | 51.6 | 12 | 48.0 | 20 | 54.1 | | |
| | often/always | 23 | 37.1 | 12 | 48.0 | 11 | 29.7 | | |
| Attended places where others used drugs | no | 56 | 90.3 | 23 | 92.0 | 33 | 89.2 | F | 1.000 |
| | yes | 6 | 9.7 | 2 | 8.0 | 4 | 10.8 | | |
| STI history | no | 40 | 64.5 | 12 | 48.0 | 28 | 75.7 | 3.86 | **0.050** |
| | yes | 22 | 35.5 | 13 | 52.0 | 9 | 24.3 | | |
| Nosocomial exposure | no | 38 | 61.3 | 12 | 48.0 | 26 | 70.3 | 2.25 | 0.134 |
| | yes | 24 | 38.7 | 13 | 52.0 | 11 | 29.7 | | |
| Accidental exposure | no | 52 | 83.9 | 22 | 88.0 | 30 | 81.1 | F | 0.726 |
| | yes | 10 | 16.1 | 3 | 12.0 | 7 | 18.9 | | |
| Sex with male PWID | no | 48 | 77.4 | 12 | 48.0 | 36 | 97.3 | 18.02 | **0.000** |
| | yes | 14 | 22.6 | 13 | 52.0 | 1 | 2.7 | | |
| Sex with MSM | no | 61 | 98.4 | 25 | 100.0 | 36 | 97.3 | F | 1.000 |
| | yes | 1 | 1.6 | | | 1 | 2.7 | | |
| Sex with male PLWH | no | 14 | 22.6 | 3 | 12.0 | 11 | 29.7 | 1.76 | 0.184 |
| | yes | 48 | 77.4 | 22 | 88.0 | 26 | 70.3 | | |
| Sex with male sex worker | no | 60 | 96.8 | 25 | 100.0 | 35 | 94.6 | F | 0.511 |

(Continued)

**Table 3.** (Continued)

| | | Total | | Partners' SMoT | | | | Chi-sq. | p-value |
|---|---|---|---|---|---|---|---|---|---|
| | | | | IDU | | Not IDU | | | |
| | | N | Col. % | N | Col. % | N | Col. % | | |
| | yes | 2 | 3.2 | | | 2 | 5.4 | | |
| Selling sex for money | no | 55 | 88.7 | 22 | 88.0 | 33 | 89.2 | F | 1.000 |
| | yes | 7 | 11.3 | 3 | 12.0 | 4 | 10.8 | | |
| History of IPV with named partners | no | 52 | 83.9 | 16 | 64.0 | 36 | 97.3 | F | **0.001** |
| | yes | 10 | 16.1 | 9 | 36.0 | 1 | 2.7 | | |

SMoT, survey-based mode of HIV transmission; IDU, injecting drug use; AGYW, adolescent girls and young women; STI, sexually transmitted infection; PWID, people who inject drugs; MSM, men who have sex with men; PLWH, people living with HIV, IPV, intimate partner violence.

F denotes that Fischer's exact test was used instead of chi-square test.

HCV testing, indicating that approximately a third of IDU-transmitted cases are misclassified as heterosexual. This estimate is nearly identical to the one obtained in the previous survey in 2013–2015, although the proportion of IDU-related cases was notably higher at that time (70% among men) [10].

The current study adds to the literature, suggesting a significant association between injecting drug use, unprotected sex, and heterosexual transmission to non-drug-injecting women. Our findings highlight the need for tailored prevention interventions for serodiscordant couples and bridge populations (non-drug-using sexual partners of PWID), considering the complex interplay between HIV risks, IPV, STI, non-disclosure of HIV and IDU, and combined IDU- and HIV-related stigma to effectively address the unique challenges faced by PWID and their partners in Ukraine and beyond [37]. Additionally, such intervention could benefit from bottom-up approaches nested around lived experiences of AGYW in Ukraine to account for unstable socio-ecological environments (i.e., the ongoing war or pandemics) [37–40].

## Limitations

Our approach to identifying IDU exposure relied on participants' self-report. Due to multiple forms of stigma [41–43], it is likely that this behavior was underreported, particularly by women [42,44]. We used HCV seropositivity as a marker of drug injection, yet we cannot entirely exclude the possibility that some women in our sample were infected through IDU, and that the proportion of PWID among Partners was underestimated. This, however, lends confidence that our bridge population estimate (40%) represents a conservative minimum.

Another important limitation pertains to the relatively small number of enrolled Partners (n = 65). The response rate among partners was lower than expected, to a significant extent due to the COVID-19 pandemic and lockdown measures. These restrictions complicated our study enrollment, affecting public transportation and clinic attendance. We refrained from artificially increasing incentives to boost recruitment, as it could jeopardize the validity of index-partner relationships. The small sample size limits the precision and generalizability of our findings.

It is possible that the ability to enroll a Partner was not randomly distributed among women, introducing the possibility of selection bias. To address this concern, we conducted a sensitivity analysis, indicating that women who recruited a Partner were more aware of HIV status of their sexual contacts. This may result from our recruitment approach that excluded partners deemed 'unsafe' in terms of IPV. Other key characteristics of women did not differ, suggesting that the extent of this bias was moderate. Moreover, the exclusion of 'unsafe'

partners further limits the generalizability of partner-related results and may contribute to the underestimation of the bridge population size.

Finally, due to the descriptive nature of the study, we did not perform in-depth analyses of causal relationships between the key independent variable (MoT of Partners) and potential outcomes. Therefore, the observed associations in both study groups can be explained by confounding due to measured or unmeasured factors.

## Conclusion

Our study revealed that at least 40% of women who acquired HIV via heterosexual mode in 2016–2019 in Ukraine had a PWID sexual partner before seroconversion. This estimate underscores the significant contribution of this bridge population to the ongoing HIV epidemic in Ukraine. Women who had PWID partners also reported a higher prevalence of STI, low condom use, and significantly greater experience of IPV. These findings emphasize the need for intensified, targeted HIV prevention efforts among PWID and their sexual partners, particularly non-IDU AGYW. The prevention interventions and index testing strategies should comprehensively address the biological and structural determinants of transmission between key and bridge populations, such as IDU- and HIV status disclosure, STIs, IPV, and stigma. The ongoing large-scale index testing programs in Ukraine may provide additional data to assess the trends in transmission dynamics in key and bridge populations.

## Supporting information

**S1 Table. Partner recruitment status.**
(DOCX)

**S2 Table. HIV risk factor definitions.**
(DOCX)

**S3 Table. Comparison of socio-demographic characteristics and HIV risk factors of AGYW, by partner enrollment status.**
(DOCX)

**S1 Data.**
(CSV)

**S2 Data.**
(CSV)

## Author Contributions

**Conceptualization:** Maryna Kornilova, Alexandra Deac, Olga Morozova, Olga Varetska, Kostyantyn Dumchev.

**Data curation:** Alexandra Deac, Kostyantyn Dumchev.

**Formal analysis:** Oleksandr Zeziulin, Kostyantyn Dumchev.

**Funding acquisition:** Oleksandr Zeziulin, Olga Varetska.

**Investigation:** Kostyantyn Dumchev.

**Methodology:** Oleksandr Zeziulin, Maryna Kornilova, Olga Varetska, Kostyantyn Dumchev.

**Project administration:** Oleksandr Zeziulin, Maryna Kornilova, Olga Varetska, Iryna Pykalo.

**Resources:** Maryna Kornilova, Olga Varetska.

**Software:** Alexandra Deac, Kostyantyn Dumchev.

**Supervision:** Alexandra Deac, Olga Morozova, Olga Varetska, Iryna Pykalo, Kostyantyn Dumchev.

**Validation:** Kostyantyn Dumchev.

**Visualization:** Alexandra Deac.

**Writing – original draft:** Oleksandr Zeziulin, Alexandra Deac, Olga Morozova, Kostyantyn Dumchev.

**Writing – review & editing:** Oleksandr Zeziulin, Alexandra Deac, Olga Morozova, Iryna Pykalo, Kostyantyn Dumchev.

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
