## [Decision Letter · Decision Letter 0]

21 Dec 2023

PONE-D-23-32276The modes of HIV transmission among young women registered in HIV clinics and their sexual partners in Ukraine.PLOS ONE

Dear Dr. Zeziulin,

Thank you for submitting your manuscript to PLOS ONE. After careful consideration, we feel that it has merit but does not fully meet PLOS ONE’s publication criteria as it currently stands. Therefore, we invite you to submit a revised version of the manuscript that addresses the points raised during the review process.

We look forward to receiving your revised manuscript

Kind regards,

Justyna Dominika Kowalska

Academic Editor

PLOS ONE

“The study was conducted under the financial support of  International Charitable Foundation "Public Health Alliance" (hereinafter the Alliance) is a leading professional organization that, in cooperation with key public organizations, the Ministry of Health and other government bodies of Ukraine, fights the HIV/AIDS epidemic in Ukraine, managing preventive programs and providing quality technical support and financial resources to organizations.

All these efforts are aimed at achieving in the country universal access to comprehensive services for HIV/AIDS, tuberculosis and viral hepatitis C in Ukraine and an effective response to the epidemic at the community level, based on the achieved results and best practices. As an independent legal entity, registered in Ukraine since 2003 and after acquiring managerial independence since January 2009, the Alliance shares the values and remains a member of the global partnership of the Alliance for Public Health (an international charitable organization that unites 30 organizations from different countries, with the Secretariat in . Hove, UK).

The Alliance's mission is to reduce the spread of HIV infection and AIDS-related mortality and reduce the negative impact of the epidemic by supporting public response to the HIV/AIDS epidemic in Ukraine, as well as by spreading effective approaches to HIV prevention and treatment in Eastern Europe and Central Asia.

The main programs currently carried out by the Alliance are:

• the "Investment in Impact on Tuberculosis and HIV" program, financed by the Global Fund to Fight AIDS, Tuberculosis and Malaria;

• the program "Improving the cascade of HIV treatment for key population groups by means of differentiated detection of new cases and involvement in treatment, building the potential of the Public Health Center of the Ministry of Health of Ukraine and strategic information in Ukraine", financed as part of the METIDA international technical assistance project;

• others.

This procurement of this study was carried out under the project "Accelerating progress in reducing the burden of tuberculosis and HIV infection by providing universal access to timely and high-quality diagnosis and treatment of tuberculosis, expansion of evidence-based prevention, diagnosis and treatment of HIV infection, creation of viable and stable health care systems ", with the support of the Global Fund.”

3. In the online submission form, you indicated that [The data set contains sensitive information on the characteristics of peaple living with HIV and will be issued upon the official request to the Alliance for Public Health, Ukraine.].

Reviewers' comments:

Reviewer's Responses to Questions

**Comments to the Author**

1. Is the manuscript technically sound, and do the data support the conclusions?

Reviewer #1: Yes

Reviewer #2: Partly

2. Has the statistical analysis been performed appropriately and rigorously

Reviewer #1: Yes

Reviewer #2: Yes

3. Have the authors made all data underlying the findings in their manuscript fully available?

Reviewer #1: Yes

Reviewer #2: Yes

4. Is the manuscript presented in an intelligible fashion and written in standard English?

Reviewer #1: Yes

Reviewer #2: Yes

5. Review Comments to the Author

Reviewer #1: The title of the manuscript is acceptable and informative, the introduction reflects the purpose of the study - to investigate the role of the bridge populations in the HIV epidemic in Ukraine. Research approach and Enrollment methodology

correspond to the study hypotheses and are sufficienly detailed: assessment was based on measuring the point prevalence of Hepatitis C and IDU as outcomes and probable predictors in the target population to calculate statistical associations using Pearson’s chi-square test.

Research results based on analysis of available data, taking into account strengths and limitations

assessible, well substantiate the Conclusion and the importance of future studies.

The References list contains the most significant publications of recent years related to the research topic

Reviewer #2: The general comments to the paper are:

The paper describes a study investigating modes of HIV transmission in adolescent girls and young women in Ukraine and their links to PWID. I believe that this study addresses important questions, especially for an Eastern European country with the high prevalence of injecting drug use and HIV. The study utilizes both a survey and serological confirmatory testing, along with obtaining information from the national HIV database. The study participants were enrolled from nine different regions of Ukraine, which is a great strength considering the size of the country and the known disparity in healthcare, HIV prevalence, and economy between the regions.

I believe this paper will be of interest to readers, and I hope the comments below can improve it and eliminate the concerns that might arise.

I think that this paper could be substantially improved by shortening the Introduction section and focusing only on the most relevant findings of the previous studies.

It was not fully clear for me as a reader that the research question is the prevalence of HCV in partners of the AGYW enrolled in the study. A substantial emphasis is put on AGYW, the section on their partners in Result is the second one, and I only found that the main question was the prevalence of HCV in partners, from Supplementary 2. The authors might consider reshaping the paper by starting the Results section from the merged Table 1 including both AGYW and their partners, then going by the order of their hypotheses as they were listed in Supplementary 3.

I would suggest to also emphasize the main research question more in Discussion. I believe there is room for comparison also with non-Eastern European studies, especially given that the authors want to stress the role of stigma in IDU underreporting and the higher barriers to accessing care in Ukraine.

I would advise the authors to check the paper against the People First Chapter terminology guidance, specifically avoiding such words as “HIV-infected” or “having HIV seropositive status” which can be replaced by “living with HIV” (https://peoplefirstcharter.org/#:~:text=The%20People%20First%20Charter%20was,language%20perpetuates%20stigma%20%26%20marginalises%20people)

I believe some check of the references is needed; in a couple of places, the information in the paper does not match the data in the references.

I also believe this paper could be slightly improved also in terms of the language and grammar; currently, there are some places where the meaning of the text is unclear, and where the language check will improve the readability.

More specific comments by section are provided in the attached document.

6. PLOS authors have the option to publish the peer review history of their article (what does this mean?). If published, this will include your full peer review and any attached files.

Reviewer #1: **Yes: **Nataliya Nizova Ph.D. M.D. Professor

Reviewer #2: No

---

## [Author Response · Author response to Decision Letter 0]

19 Mar 2024

We thank the Reviewers and Editors for a thoughtful and detailed review and, agreeing with the comments, have revised the manuscript significantly.

1. We revised financial disclosure.

2. We included study data sets as supplementary information.

3. We included full ethics statement in the ‘Methods’ section of the manuscript

---

## [Decision Letter · Decision Letter 1]

23 May 2024

Modes of HIV transmission among young women and their sexual partners in Ukraine.

PONE-D-23-32276R1

Dear Dr. Oleksandr Zeziulin,

We’re pleased to inform you that your manuscript has been judged scientifically suitable for publication and will be formally accepted for publication once it meets all outstanding technical requirements.

Kind regards,

Justyna Dominika Kowalska

Academic Editor

PLOS ONE

Additional Editor Comments (optional):

Reviewers' comments:

Reviewer's Responses to Questions

**Comments to the Author**

1. If the authors have adequately addressed your comments raised in a previous round of review and you feel that this manuscript is now acceptable for publication, you may indicate that here to bypass the “Comments to the Author” section, enter your conflict of interest statement in the “Confidential to Editor” section, and submit your "Accept" recommendation.

Reviewer #1: All comments have been addressed

2. Is the manuscript technically sound, and do the data support the conclusions?

Reviewer #1: Yes

3. Has the statistical analysis been performed appropriately and rigorously? 

Reviewer #1: Yes

4. Have the authors made all data underlying the findings in their manuscript fully available?

Reviewer #1: Yes

5. Is the manuscript presented in an intelligible fashion and written in standard English?

Reviewer #1: Yes

6. Review Comments to the Author

Reviewer #1: The authors carefully and creatively worked on the comments, which seriously strengthened the evidence of the research hypothesis and the validity of the conclusions.

7. PLOS authors have the option to publish the peer review history of their article (what does this mean?). If published, this will include your full peer review and any attached files.

Reviewer #1: **Yes: **Nataliya Nizova

---

## [Editor Report · Acceptance letter]

18 Jun 2024

PONE-D-23-32276R1 

PLOS ONE

Dear Dr. Zeziulin, 

I'm pleased to inform you that your manuscript has been deemed suitable for publication in PLOS ONE. Congratulations! Your manuscript is now being handed over to our production team.

Kind regards, 

on behalf of

Prof. Justyna Dominika Kowalska 

Academic Editor

PLOS ONE